# Notes on Developing Research Review in Urban Planning and Urban Design Based on PRISMA Statement

Hisham Abusaada [1] and Abeer Elshater [2,*]

1 Housing and Building National Research Center, Cairo 12622, Egypt
2 Department of Urban Design and Planning, Faculty of Engineering, Ain Shams University, Cairo 11566, Egypt
* Correspondence: abeer.elshater@eng.asu.edu.eg

**Abstract:** The point of view expressed in this article is theoretically grounded in the PRISMA statement, which is a tool for critically evaluating academic papers in public health. Bibliometrics analysis, systematic review, meta-analysis, and storytelling techniques (BSMS) were used to identify relevant studies and create a process for documented urban planning and design research. To promote the construction of new facts based on compelling evidence reported in earlier literature reviews, academics in urban planning and urban design are encouraged to build their own suitable review procedures to support the formation of conclusions based on compelling evidence. Providing a strategic approach and practice process is one of the significant contributions of this knowledge research.

**Keywords:** bibliometric analysis; systematic review; meta-analysis; storytelling techniques

## 1. Introduction

A group of public health academics developed the Preferred Reporting Items for Systematic Reviews and Meta-Analyses (PRISMA) statement (Moher et al. 2009). Since 2009, PRISMA has been used in academic studies in the medicine and animal sciences (Fleming et al. 2014; Purssell and McCrae 2020; Abusaada and Elshater 2021a) and nursing (Sandelowski et al. 2007). Page et al. (2021) have updated the PRISMA statement for reporting systematic reviews. Some urban planning (UP) and urban design (UD) academic publications have used this statement in their public health literature, explaining its elements and how this agreement can be registered with the National Institute for Health Research in the UK. The PRISMA statement still does not appear to meet the criteria for its use in UP and UD studies, even after the use of a structured, systematic review and meta-analysis.

A systematic review is not new for UP and UD, as it has been used for years to examine the themes of urban environments in much of the academic literature (Francini et al. 2021; Elshater and Abusaada 2022; Abusaada and Elshater 2021b). City planners have developed experimental scientific methods and methods that fit the requirements of their theoretical and experimental research (MacCallum et al. 2019) using a quantitative bibliometric analysis approach to review the literature (Verweij and Trell 2019). Others have developed quantitative research methods (Ewing and Park 2020; Abusaada and Elshater 2021c) and analysis-of-variance (ANOVA) methods (Stoker et al. 2020), which they used to prove and verify the validity and reliability of research sources (Duke et al. 2020). Finally, a combination of systematic reviews and meta-analyses emerged in the medical sciences (Moher et al. 2009), which is used in the same way in urban research (Liu and Niyogi 2019).

UP and UD have yet to benefit from advances in integrated social and medical science methods in the literature review. According to Xiao and Watson (2019), the field of planned action research lacks systematic reviews. Because of a lack of systematic reviews in UD and UD, the readership of such publications often asks, how can PRISMA statement be applied in UD and UP research?' This was encountered when the authors attempted

to publish several papers that had undergone rigorous peer review. Considering these personal experiences, it is believed that the present study design is necessary. Therefore, it was decided to build an instrument for conducting systematic reviews that would be trustworthy through collaboration with medical reviewers. For the second stage of the study, the focus was on establishing a systematic method that makes it possible to increase knowledge of accessible sources and utilize these sources to examine, analyzes, and synthesize the information.

Therefore, with this viewpoint, an attempt is made to map out how to present a literature review about the patterns, approaches, and methodology of data mining. The findings of this study should help novice researchers learn how to conduct an integrated literature review. In addition, the benefits can be of interest to early career reviewers and evaluators to gain a better understanding of the potential benefits and drawbacks of various literature review approaches and components, as outlined in the article. With this viewpoint, the bibliometric, systematic, meta-analysis, and storytelling (BSMS) processes are compiled to be adapted in UP and UD research. In a manner similar to the PRISMA statement protocols, UP and UD researchers need to compile a database or platform for published research.

## 2. Prior Studies

The use of academically applied bibliometric analysis (Amirbagheri et al. 2019), content analysis (Neuendorf 2016), and snowball techniques (Dastjerdi et al. 2021; McLeod and Schapper 2020) within scientific research methods has seen a significant increase in many areas of specialization, such as projection-based clustering through self-organization and swarm intelligence (Thrun 2018), and qualitative research methods in language (Tracy 2020). The systematic review has been used as a technique in the social sciences (Boland et al. 2014), medical sciences (Higgins and Green 2019) and nursing (Purssell and McCrae 2020). Nevertheless, it still has a clear and unambiguous definition (Moher et al. 2009).

According to Ball (2017), researchers have adopted the quantitative bibliometric technique to review the literature, while Verweij and Trell (2019) and Tracy (2020) noted that other researchers use qualitative comparative analysis (QCA) and its multiple methods for systematic and integrated literature reviews. In this regard, scholars have developed quantitative research methods (QRM) (Ewing and Park 2020) and analysis of variance methodologies (ANOVA) (Stoker et al. 2020), which they utilized to establish the validity and trustworthiness of research sources (Duke et al. 2020). There is also a combination of systematic reviews and meta-analyses in the medical sciences (Liberati et al. 2009; Moher et al. 2009; Noordzij et al. 2011) and urban research (Liu and Niyogi 2019; Smith et al. 2021). A group of public health academics used the PRISMA statement as a set of preferred reporting items for systematic reviews and meta-analyses (Khan et al. 2011; Moher et al. 2009). This has been replicated in academic research in medicine (Fleming et al. 2014; Liberati et al. 2009; Purssell and McCrae 2020) and nursing (Sandelowski et al. 2007) since 2009, and explored in the public health literature (Page et al. 2021). The National Institute for Health Research (NIHC) registered PRISMA statement in public health, and some urban planning and design academics have used it in their literature (Smith et al. 2021).

According to a review of healthcare research, the PRISMA statement's goal is to help researchers improve the writing of reports that are relevant to structuring systematic reviews and meta-analyses. After the inclusion and exclusion phases, the remaining references that were analyzed. The PRISMA statement is based on a methodology that explains why a review was undertaken and what the authors did and discovered in a transparent manner (Liberati et al. 2009). Page et al. (2021) developed methods for identifying, selecting, evaluating, and synthesizing studies. The treaty now explains the sequence of systematic reviews organized across 27 items such as: (1) Reference title; (2) Summary; (3–4) Introduction (logic and objectives); (5–15) Methodology (eligibility criteria, information sources, search strategy, information base, selection process, information collection process, listing, and definition of the variables for which data were sought or grants, risk assessment

of bias, summary of measurements, statistical or meta-analysis synthesis of results, bias assessment, and certainty assessment); (16–22) Outcomes (case selection, characteristics, case study bias, individual cases, outcome synthesis, cross-case bias, and additional analyses); (23) Discussion (summarizing evidence, limitations, and summary); and (24–27) other information.

Academics have used the PRISMA statement in the urban studies (Smith et al. 2021). The PRISMA statement states these studies consisted of the following elements: title, authors' names, and citation method; the research questions and what the research aims to establish; the information base on which the study demonstrates the inclusion and exclusion criteria (and independent reviewers from outside the research team can participate in this stage). The state and field of study include the participants (i.e., individuals, groups, municipalities), intervention(s) (i.e., focus on survey forms), exposure(s), comparison (i.e., focus on cases that are not selected), and comparator(s)/control, biosphere context, quantitative and qualitative outcome data. Bias (quality) risk and assessment strategy for data syntheses include narrative synthesis of the available evidence, scoring by reviewers, and method of review (narrative and structured methodology). Finally, the date of the report, termination, funding, conflict of interest, language, country, and acknowledgement of modernization are all included in a body called PROSPERO.

However, PRISMA still does not meet the criteria for its use in urban planning and design studies. According to Moher et al. (2009), the current research is in itself building its main problem; although the preparation of widely used systematic review reports is currently in progress, their quality is still inconsistent. Fleming et al. (2014) added that PRISMA reports that are consciously used in medical studies do not use meta-analysis of observational studies in epidemiology (MOOSE) and the quality assessment of the accuracy of diagnostic studies (QUADAS).

## 3. Causal and Speculation Arguments

Based on experience with peer-reviewed journal publications, the authors developed this viewpoint article. Throughout the past two years, the authors have reviewed several peer-reviewed research articles in prominent databases such as Web of Science and Scopus. The articles were examined for their research question commitment along four axes: strength of the arguments, originality of the arguments, the article's impartiality and its integrity (the article's adherence to governing restrictions). The ability of the article to meet the requirements for applying research methods such as the depth of results, discussion, and commitment to writing style and readability, were also evaluated.

In many systematic literature reviews, a simple query outlines the purpose and issue of the study topic according to the findings. There were several studies that did not ask any questions in environmental management, medical sciences, or UP. Despite this, the findings suggest that having more than one question is not always a bad thing, particularly, if the questions are interdependent and help to provide a broader picture of a research subject. Questions that go beyond a simple literature review must be separated out, so that a clear research topic for the review itself can be identified and stated in the methodology section where the research methodologies and procedures are discussed in detail.

The hypothesis in the article must be supported by different theories, concepts, or ideas based on theoretical findings. Furthermore, it is necessary to conduct a critical analysis of the literature that has been classified as sources of information and draw conclusions that can be used in future studies, both from researchers supporting and opposing the subject of the research or from experts, practitioners, and decision-makers who support or oppose it. A high degree of awareness and rigor should be used while conducting a literature review, explaining the situations and reasons and the evidence for each of them. Moreover, the literature review should conclude with sufficient new information to constitute 'originality' and 'contribution'.

Electronic databases such as the Web of Science, Scopus, and Elsevier, were used to assess the dependability of a journal. Scores in these databases are recognized to be based

on the number of citations in books, journals, and other scientific publications. The question of reliability is still under study by researchers, and thus far, there is no consensus that any bibliometric index is better than the *h* index (Hirsch 2010).

When reviewing research relevant to UP and UD, it was found that many researchers used nomenclature for research methods in the main title, abstract, and methodology section without fulfilling the PRISMA application requirements. To reflect the PRISMA Statement, research design requires the preparation of an agreement of 27 elements and that this agreement be registered in a specific data entity called The International Prospective Register of Systematic Reviews (PROSPERO). Fleming et al. (2014) argued that this convention is still used while ignoring many other relevant guidelines, negatively affecting the results of systematic reviews. In addition, some researchers mentioned their use of meta-analysis without fulfilling the requirements to present statistics or provide a satisfactory descriptive analysis of the results (Chapain and Sagot-Duvauroux 2020). The applications of PROSPERO database were used to investigate the differences between those academic studies and others that used the meta-analysis (Liu and Niyogi 2019).

An analytical examination of additional and specialized research demonstrated that the theoretical component of most research agreed with what Fleming et al. (2014) proposed about the research sources, as follows. It is important to use strategies that are objective while searching for the best sources without bias. Diverse sources of information make the discovery of new findings possible. A thorough comprehension of the relation of the source to the issue of the study is required. Because of their dependence on the Cochrane library/review/database (Higgins and Green 2019), it was observed that public health research findings are more organized than those obtained in UP and UD databases. Therefore, urban planners and urban designers are encouraged to implement such a database. To build this database, a team of researchers from diverse fields must work together. Furthermore, this research should have appropriate time to launch an UP and UD information base.

## 4. BSMS Process

With the above viewpoint, a statement of research design in UP and UD is proposed. It follows the Statement of Preferred Reporting Elements components in the PRISMA structured methodology and meta-analyses. The process combines bibliometric analysis, systematic review, meta-analysis, and data-driven storytelling techniques. It is called the 'BSMS process in UP and UD', which includes five stages, as shown in Figure 1.

### 4.1. First Stage: Framing the Topic and Its Importance—Identifying the Gaps

This stage tests the extent to which UP and UD require a new theoretical study, which includes four steps:

1. Identifying the scope of the investigation: subject area and category.
2. Determining a group of words about the proposed research, which could be found using the back-and-forth snowball technique, and after a lengthy reading in the previously available publications, dealing with the research topic.
3. Determining the source types: book, chapter in a book, refereed journal, grey literature and so forth.
4. Exploring the repetition of keywords in UP and UD literature through a statistical study of their frequencies and their relationship to the words that appear with them constantly in the title, abstract, and index words, as expressed in a graphical abstract using the concept of the green supply chain and using VOSviewer (this step identifies the gaps in the subject of the study on which the researcher will work) (Elshater et al. 2022).

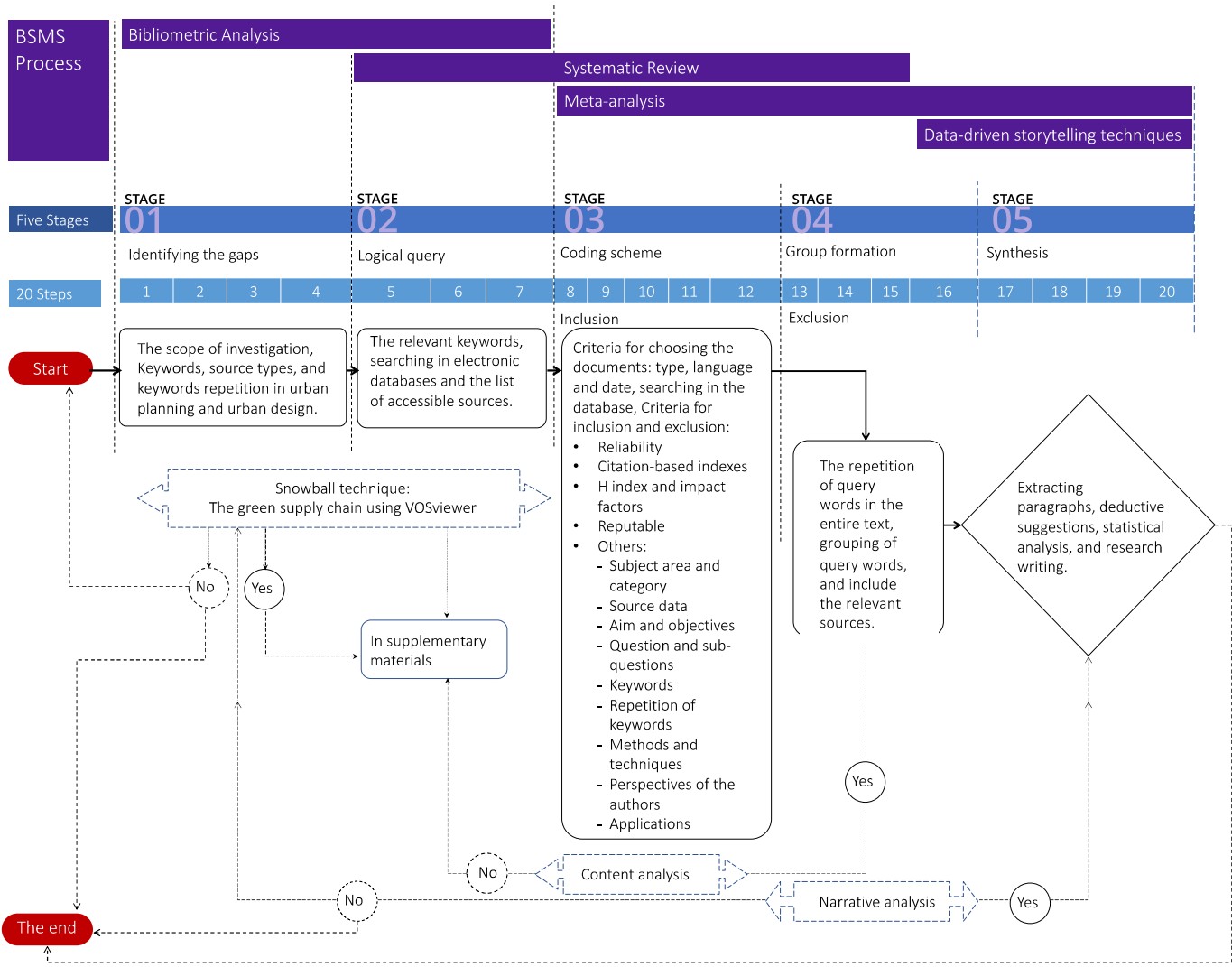

**Figure 1.** BSMS process in UP and UD.

### 4.2. Second Stage: Logical Query—Available Resources

The research sources are identified using bibliometric analysis in this step. As part of their study, researchers choose specific words or terms that they want to search for in the databases. There are three steps in this stage:

5.  Choosing the relevant keywords used in the study and deleting those not closely related to the research work.
6.  Searching in electronic databases for the selected keywords using three logical operators in the case of searching for one word and using 'OR' or 'AND' when searching for two different words and using 'NOT' to exclude a word that you do not want to search.
7.  Writing down the list of accessible sources, without excluding a source for any reason if it includes the subject area, domain category, and keywords.

### 4.3. Third Stage: Inclusion and Exclusion—Coding Scheme

This stage begins with preparing the initial inclusion and exclusion requirements for the sources on which the research depends, such as a book, a chapter in a book, a study in a journal, a conference, reports, and blogs. The first step of the coding scheme stage passes by setting the exclusion criteria: search language, publication year, word frequency, and query

being in the title, abstract, and main text. The include/exclude stage uses a methodological filter and includes three steps:

8.   It is important to define the selection criteria for each document if it is any kind of written work, whether it is a whole book or part of a book; documents in scientific or technical reports; letters and correspondence between parties and individuals; elite speeches; and websites. These are examples of grey literature or secondary data. Regarding sources, there is a broad variety of alternatives. Some research uses only English sources, while others incorporate translations into English as well as translations from other languages. The date of publication of a book is also included.

9.   Criteria for searching the documents by defining the subject area and field category using electronically registered databases such as the Web of Science database of journal citation research (JCR) and the Scimago Journal & Country Rank (SJR) database supported by Scopus.

10.   Criteria for inclusion and exclusion in the study are used. The credibility of a published document is determined by a peer-review process. Citation-based indices such as Web of Science, Scopus, and Elsevier, are included. According to the indicator's value and the measurement impact factor, journals are included or excluded. The *h* index is assigned to the study or the researcher. A respected academic publication is included.

11.   Other criteria specific to each area of research are used to achieve comprehensiveness or reduce conflict between specializations:

-   Research areas of specialization: subject area and category;
-   Source data (date, place of publication, impact factor, and document title);
-   Main aim and objectives;
-   Main questions and sub-questions;
-   Keywords in the title, abstract, and indexing words
-   Repetition of keywords in the title, abstract, index words, and main text;
-   Methods and techniques: individual or combined, and theoretical or experimental;
-   Perspectives of the authors (select quotes from the text);
-   Extent of commitment to applying the requirements of the subject applications (theoretical and experimental); and
-   Source name.

12.   For any study, it is possible to add supplementary materials which contain the data for the books by arranging the sources by their categorization: International Standard Book Number (ISBN), publisher, date of publication, book title, the title of a chapter in a book and its abstract, number of book citations and individual author citations, source name and location, and the number of references to each author in a book.

*4.4. Fourth Stage: Reading Texts—Content Analysis and Group Formation*

At this stage, the content analysis approach is used to read the entire text of the manuscripts selected for investigation. The second stage of inclusion and exclusion is based on the evidence from the relevance of the published content. Sources of information are excluded that do not provide 'useful evidence to answer the questions of the current study', while relying on the critical element of retaining the sources that contain helpful information through the principle of 'reducing data to manageable representations' (Neuendorf 2016) which includes four steps:

13.   Determining the query words and identifying their repetition in the entire text.
14.   Categorizing keywords into separate groups followed by sub-words.
15.   Excluding sources that do not include these groups or irrelevant words to sources.
16.   Including the relevant sources.

*4.5. Fifth Stage: Reading Texts—Synthesis of Results*

In this stage, data-driven storytelling techniques are used, ranging from descriptive statistics to inferential statistics. Systematic review, narrative synthesis, and meta-analysis approaches are used, which include four steps:

17. Extraction of whole paragraphs (or assertions) from the study's literature and placement in the source table after a thorough review.
18. Between these paragraphs, a narrative synthesis that delivers deductive suggestions appropriate for the new investigation is extracted.
19. As a result of the meta-analysis (descriptive or meta-analysis), gaining the ability to determine how many times certain sections repeat and how much inferential evidence exists to support the findings of the current study (statistical analysis).
20. Writing the research in a way that is both readable and understandable.

## 5. Conclusions

Scientific writing techniques and procedures in UP and UD still require greater attention to produce data and standards for essential research issues. Understandings of this sort are documented at medical facilities under the PRISMA statement or agreement. The development of the BSMS process for UP and UD might lead to reference registration information centers. This process relies on bibliometric analysis, systematic review, meta-analysis, and data-driven storytelling methodologies to conduct research. Further research should concentrate on detecting patterns and causal linkages, evaluating them across different studies or instances, and designing methods for conducting research. Other strategies are utilized to obtain credible sources and to devise forms and procedures for developing questions. Providing an approach and practice process is one of the significant contributions of this knowledge research. This contribution opens a dialogue by presenting the implications of the results concerning the context of research, joint issues, interventions, constants and variables, and theory and practice experiences. Urban planning and design researchers can use this approach to analyze and document the literature used in previous works.

**Author Contributions:** All authors collaborated on the conceptualization, methodology, writing review, and editing of the manuscript. All authors have read and agreed to the published version of the manuscript.

**Funding:** The study was supported by a grant from The Science, Technology, and Innovation Funding Authority (STDF) under grant number [STDF-BARG 37234].

**Acknowledgments:** It is with great appreciation that the authors would like to thank the editors and reviewers for their constructive comments.

**Conflicts of Interest:** The authors declare no conflict of interest. The authors also declare that the research funder had no role in the design of the study; in the collection, analyses, or interpretation of data; in the writing of the manuscript, or in the decision to publish the results.

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
