# Peer review of "Notes on Developing Research Review in Urban Planning and Urban Design Based on PRISMA Statement"

_socsci, doi:10.3390/socsci11090391_

Round 1

Reviewer 1 Report

I find this manuscript meets the criteria for publication, however there are a few minor mistakes that should be corrected such as:

- Line 200 is misplaced (First stage title), it should be moved to line 185.

- The steps from second stage onwards should start from 5 instead of starting from 1 again, in order to be coherent with Figure 1.

The short length and the number of references are acceptable for a Viewpoint manuscript, however I encourage you to continue with your research on this topic in the future to obtain further useful conclusions!

Author Response

I find this manuscript meets the criteria for publication, however there are a few minor mistakes that should be corrected such as:

Response: Thank you so much for your supportive feedback. 

- Line 200 is misplaced (First stage title); it should be moved to line 185.

Response: We have moved this line based on you recommendation 

- The steps from second stage onwards should start from 5 instead of starting from 1 again, in order to be coherent with Figure 1.

Response: We arranged the steps to be matched with Figure 1, starting from 1 to 20. 

The short length and the number of references are acceptable for a Viewpoint manuscript, however I encourage you to continue with your research on this topic in the future to obtain further useful conclusions!

Response: Thank you so much for this valuable comment. We will plan to continue this viewpoint in another full-length article.  

Reviewer 2 Report

This paper, titled “Notes on developing research design in urban planning and urban design based on PRISMA statement” has been submitted as a Viewpoint. There is no mention of this category of submission in the journal Social Sciences’ Instructions to Authors but I note that reviews to the journal must follow the PRISMA guidelines. Consequently, this paper will be of interest to the journal’s readers as it explores the task of using the PRISMA statement for reviews in urban design and urban planning.

However, the paper needs some revision before it can be accepted for publication. I found the argument of the paper hard to follow, perhaps because the writing is not always clear. There is inconsistent use of the term PRISMA statement; sometimes it is referred to as “the agreement” or as “PRISMA Statement/Agreement”. This is confusing for the reader. For example, at line 19, PRISMA “statement” is given and then referred to, I think (this is not clear) at line 24 as “this agreement”. At line 25, the sentence is introduced with “Thus” but it is not obvious to me that this final sentence flows logically from the foregoing. In other places, e.g. line 50, a word such as “its” is used, but it is not obvious to what this refers. I think the authors need to reread the manuscript very carefully and clarify ambiguities such as these. There are many. I have noted these below, and other points for clarification:

·         Line 2: I think the title should be changed to reflect its focus on reviews in UD and UP (not research design) based on the PRISMA statement.

·         Line 40: who is asking ‘What questions are you aiming to answer’? It seems to be “peer-reviewed scientific publications” but this doesn’t make sense. Do you mean reviewers of such publications?

·         Line 50: this sentence is unclear to me. How does this paper “provide insights into how popular, rare……are developed”? I thought it is explaining how PRISMA can be applied in UD and UP research.

·         Line 66: What does “this systematic review” refer to here?

·         Lines 89-90 make no sense.

·         Line 101: (14-27) should read (24-27)

·         Line 151: What guidelines are you referring to?

·         Line 157: Should ‘PRISMA’ be inserted before “application”.

·         Line 185: I think line 200 should be inserted here.

·         Lines 220-231: These three items should be numbered 1-3, not 4-6.

·         Line 236: The number ‘7’ should be deleted.

·         Line 249: The number ‘8’ should be deleted.

·         Line 255: What do you mean by “the remaining texts”? What are these texts?

·         Lines 261-265: These items should be numbered 1-4.

·         Line 269: There are four steps given, not five. These should be numbered appropriately (i.e. 1-4).

·         In Figure 1, what does “excelled sources” mean?

Three references are cited but not included in the References at the end of the paper:

·         Stoker et al. (2020)

·         Liberati et al. (2009)

·         Ewing and Park (2020)

Is the date 2009 or 2007 for the reference Moher et al?

If there are only two authors in Higgins et al. (2019), both names should be given rather than ‘et al.’.

Author Response

This paper, titled “Notes on developing research design in urban planning and urban design based on PRISMA statement” has been submitted as a Viewpoint. There is no mention of this category of submission in the journal Social Sciences’ Instructions to Authors but I note that reviews to the journal must follow the PRISMA guidelines. Consequently, this paper will be of interest to the journal’s readers as it explores the task of using the PRISMA statement for reviews in urban design and urban planning.

Response: Thank you so much for your comment and feedback, which helped us to develop this version. 

However, the paper needs some revision before it can be accepted for publication. I found the argument of the paper hard to follow, perhaps because the writing is not always clear. There is inconsistent use of the term PRISMA statement; sometimes it is referred to as “the agreement” or as “PRISMA Statement/Agreement”. This is confusing for the reader. For example, at line 19, PRISMA “statement” is given and then referred to, I think (this is not clear) at line 24 as “this agreement”. At line 25, the sentence is introduced with “Thus” but it is not obvious to me that this final sentence flows logically from the foregoing.

Response: Please accept my sincere thanks for your feedback. As a matter of consistency, we have changed the description of PRISMA to 'PRISMA statement'. We also remove 'Thus'

In other places, e.g. line 50, a word such as “its” is used, but it is not obvious to what this refers. I think the authors need to reread the manuscript very carefully and clarify ambiguities such as these. There are many. I have noted these below, and other points for clarification:

Response: We have double-checked the entire text. Kindly check and let me know if further modification is needed. 

  • Line 2: I think the title should be changed to reflect its focus on reviews in UD and UP (not research design) based on the PRISMA statement.

Response: In response to your recommendation, we have modified the title. 

  • Line 40: who is asking ‘What questions are you aiming to answer’? It seems to be “peer-reviewed scientific publications” but this doesn’t make sense. Do you mean reviewers of such publications?

Response: Thank you so much for your valuable comment. We have modified the research question.  

  • Line 50: this sentence is unclear to me. How does this paper “provide insights into how popular, rare……are developed”? I thought it is explaining how PRISMA can be applied in UD and UP research.

Response: Yes, this sentence could be outside the scope. So, we have removed it. 

  • Line 66: What does “this systematic review” refer to here?

Response: We have adjusted this phase to be 'The systematic review has been ...' Please double-check Line 61.  

  • Lines 89-90 make no sense.

Response: Thank you for your comment. We deleted this unclrat sentence. 

  • Line 101: (14-27) should read (24-27)

Response: We have modified this typo. 

  • Line 151: What guidelines are you referring to?

Response: We have modifies guidelines to be  "Scores in these databases are ..." 

  • Line 157: Should ‘PRISMA’ be inserted before “application”.

Response: We have modified the text and added 'PRISMA'

  • Line 185: I think line 200 should be inserted here.

Response: We have adjusted the text for better readability. 

  • Lines 220-231: These three items should be numbered 1-3, not 4-6.

Response: We have made double-checked the numbering of our steps to keep the consistency with the text. 

  • Line 236: The number ‘7’ should be deleted.

Response: We have double-checked the numbering of our steps to keep the text consistent. 

  • Line 249: The number ‘8’ should be deleted.

Response: We have deleted this line based on your recommendation. 

  • Line 255: What do you mean by “the remaining texts”? What are these texts?

Response: We have adjusted the phrase and made it the 'entire text.'

  • Lines 261-265: These items should be numbered 1-4.

Response: We have double-checked the numbering of our steps to keep the text consistent. To clarify, we have 5 Stages, and each stage has steps. 

  • Line 269: There are four steps given, not five. These should be numbered appropriately (i.e. 1-4).

Response: We have adjusted this typography.  

  • In Figure 1, what does “excelled sources” mean?

Three references are cited but not included in the References at the end of the paper:

  • Stoker et al. (2020)
  • Liberati et al. (2009)
  • Ewing and Park (2020)

Is the date 2009 or 2007 for the reference Moher et al?

If there are only two authors in Higgins et al. (2019), both names should be given rather than ‘et al.’.